# Insights on Ferroptosis and Colorectal Cancer: Progress and Updates

**DOI:** 10.3390/molecules28010243

**Published:** 2022-12-28

**Authors:** Bangli Hu, Yixin Yin, Siqi Li, Xianwen Guo

**Affiliations:** 1Department of Research, Guangxi Medical University Cancer Hospital, Nanning 530021, China; 2Department of Gastroenterology, The People’s Hospital of Guangxi Zhuang Autonomous Region, Taoyuan Road No.6, Nanning 530021, China

**Keywords:** ferroptosis, colorectal cancer, molecular, compounds, nanoparticles, signature

## Abstract

Patients with advanced-stage or treatment-resistant colorectal cancer (CRC) benefit less from traditional therapies; hence, new therapeutic strategies may help improve the treatment response and prognosis of these patients. Ferroptosis is an iron-dependent type of regulated cell death characterized by the accumulation of lipid reactive oxygen species (ROS), distinct from other types of regulated cell death. CRC cells, especially those with drug-resistant properties, are characterized by high iron levels and ROS. This indicates that the induction of ferroptosis in these cells may become a new therapeutic approach for CRC, particularly for eradicating CRC resistant to traditional therapies. Recent studies have demonstrated the mechanisms and pathways that trigger or inhibit ferroptosis in CRC, and many regulatory molecules and pathways have been identified. Here, we review the current research progress on the mechanism of ferroptosis, new molecules that mediate ferroptosis, including coding and non-coding RNA; novel inducers and inhibitors of ferroptosis, which are mainly small-molecule compounds; and newly designed nanoparticles that increase the sensitivity of cells to ferroptosis. Finally, the gene signatures and clusters that have predictive value on CRC are summarized.

## 1. Introduction

Colorectal cancer (CRC) is the third most common cancer and the fourth leading cause of cancer-related deaths worldwide [1,2]. Its incidence is increasing in China [3], according to a recent report released by GLOBOCAN 2020. Current treatment strategies for CRC include surgery, chemotherapy, radiotherapy, immunotherapy, and targeted therapies. These treatment strategies greatly improve the prognosis of patients with CRC, especially those with early-stage disease. However, a large proportion of patients with advanced CRC benefit marginally from these treatments [4]. The most common reason for treatment failure is treatment resistance, often due to defects in cell death mechanisms [5,6]. During the past decades, many efforts have been made to overcome treatment resistance by triggering cancer cell death, including the induction of autophagy [7], pyroptosis [8], and necrosis [9] using various pharmacological or genetic interventions. Nevertheless, owing to the complex regulation of cancer cell proliferation, more treatment strategies still need to be developed to improve the treatment effect in patients with CRC.

Ferroptosis is a novel type of regulated cell death characterized by increased intracellular iron levels, reactive oxygen species (ROS), and lipid peroxidation, with genetic, biochemical, and morphological features different from those of apoptosis, unregulated necrosis, and pyroptosis (Figure 1). Therefore, ferroptosis cannot be inhibited by apoptosis, necrosis, or pyroptosis inhibitors [10,11]. The discovery of ferroptosis provides a new therapeutic strategy for overcoming drug resistance in cancer cells [12,13]. Considering the high levels of iron and ROS in cancer cells, the induction of ferroptosis in these cells may be an alternative treatment strategy for various cancers [14,15]. Recent studies have shown that ferroptosis inducers, either as a monotherapy or in combination with other chemotherapeutic drugs, can induce ferroptosis in cancer cells, especially drug-resistant cancer cells [16,17]. In addition, ferroptosis can selectively target aggressive cancer stem cells and is expected to enhance the efficacy of and overcome resistance to immunotherapy [18,19]. Therefore, ferroptosis induction may be a promising antitumor strategy for CRC, especially in patients at an advanced stage or with treatment resistance.

Considering the great potential of ferroptosis in CRC therapy and the growing number of recent studies that have reported the effect of ferroptosis in CRC, it is necessary to summarize the latest work and track the progress in this field. In this review, we first summarized the regulatory mechanism of ferroptosis, enumerated the genes that mediate ferroptosis in CRC, and introduced the action of ferroptosis inducers (including small molecules and nanomaterials) in CRC. Finally, we discussed gene signatures or clusters that could predict the prognosis of CRC to highlight the clinical translation potential of ferroptosis in CRC treatment.

## 2. Summary of the Regulatory Pathways Associated with Ferroptosis

Iron is an essential component of many enzymes involved in the normal physiological function of cells. Excess-free iron accumulation in the cells is indispensable for the occurrence of ferroptosis. Iron promotes lipid peroxidation during ferroptosis, which is considered through two pathways. The first is the production of ROS via the iron-dependent Fenton reaction (intracellular free iron and H_2_O_2_ serve as substrates), and the second is the activation of iron-containing enzymes (such as lipoxygenases). Any metabolic pathway that causes intracellular ROS accumulation and glutathione depletion can regulate ferroptosis, including iron metabolism, glutathione metabolism, and lipid peroxidation [20]. GPX4 acts as a central regulator of ferroptosis by inhibiting lipid peroxidation, and almost all regulatory pathways that modulate GPX4 levels eventually regulate ferroptosis [13]. GPX4 activity can be directly or indirectly inhibited. Erastin and RSL3 are classic GPX4 inhibitors. Erastin reduces the expression of intracellular glutathione synthesis precursors and indirectly inhibits GPX4. In contrast, RSL3 directly inhibits GPX4 activity, resulting in lipid peroxide accumulation and triggering ferroptosis [21]. 

Three main regulatory pathways govern ferroptosis development: the iron metabolic pathway, system Xc^–^/GSH/GPX4 pathway, and lipid metabolic pathway. Ferroptosis depends on high intracellular iron levels. Redox-active iron pools can directly catalyze the formation of lipid peroxides to generate destructive free radicals via the Fenton reaction, iron-dependent Fenton reaction, and the activation of iron-containing enzymes. The DNMT-1/NCOA4 [22,23], Keap1/Nrf2 [24,25], OTUD1/IREB2 [26], and Nrf2/HO-1 pathways [27,28] regulate iron levels and trigger ferroptosis. The inhibition of intracellular glutathione synthesis or blockade of glutathione-dependent GPX4 function also induces ferroptosis. Erastin inhibits system Xc^–^ (a cystine/glutamate antiporter system) on the cell membrane, resulting in a reduction in intracellular cystine intake, a reduction in glutathione synthesis, and the induction of ferroptosis [29,30]. Ferroptosis suppressor protein 1 (FSP1) can inhibit lipid peroxidation and ferroptosis by directly eliminating lipid ROS independent of GPX4 [31]. Furthermore, a massive accumulation of lipid ROS directly triggers ferroptosis, a process that can be prevented by lipophilic antioxidants and iron chelators. NADPH oxidases (NOXs) provide abundant ROS during erastin-induced ferroptosis. In addition to NOXs, membrane lipid peroxidation products are another source of ROS production that drives ferroptosis [32,33]. Polyunsaturated fatty acids (PUFAs) are the main peroxidation substrates for ferroptosis in cell membranes. Increasing PUFA synthesis enhances the sensitivity to ferroptosis, which is positively regulated by ACSL4 [34]. Furthermore, ESCRT-III is essential for the formation and abscission of intraluminal endosomal vesicles. The ESCRT-III pathway is activated during ferroptosis to separate damaged membranes in cancer cells [35]. Apart from *p53* [36], *LPCAT3* [37] and 15-*LOX*/*PEBP1* [38] also regulate ferroptosis via different pathways (Figure 2). 

## 3. Molecules That Mediate Ferroptosis in CRC

### 3.1. Ferroptosis-Inhibiting Genes

Ferroptosis is regulated by many oncoproteins, tumor suppressor genes, and oncogenic signal transduction pathways. Some genes promote ferroptosis, while others inhibit ferroptosis in CRC via distinct pathways. Most genes that inhibit ferroptosis are expressed when GPX4 levels decrease. *SFRS9* expression was shown to be elevated in CRC tissues compared with paracancerous tissues. The overexpression of *SFRS9* inhibits CRC cell death and lipid peroxidation induced by erastin and sorafenib, and the inhibitory effects of ferroptosis are mediated by decreased GPX4 levels [39]. *HSPA5* repressed ferroptosis to promote CRC development by maintaining GPX4 stability, which slowed GPX4 degradation to give CRC cells more time to adjust to erastin toxicity [40]. The knockdown of *TIGAR* significantly increased erastin-induced ferroptosis in CRC cells, decreasing the GSH/GSSG ratio and increasing lipid peroxide production. In addition, the inhibition of *TIGAR* repressed *SCD1* expression in a redox- and AMPK-dependent manner, suggesting that *TIGAR* induces ferroptosis resistance in CRC cells via the ROS/AMPK/SCD1 pathway [41]. The knockout of *SLC7A11* increased ROS levels and reduced cysteine and glutathione levels in CRC stem cells, which reduced GPX4 levels and resulted in lipid peroxidation [42]. *TP53* limits ferroptosis by blocking *DPP4* activity in a transcription-independent manner. The loss of *TP53* prevents the nuclear accumulation of *DPP4*, facilitates lipid peroxidation, and subsequently results in ferroptosis [43]. 

### 3.2. Ferroptosis-Promoting Genes

Some genes promote ferroptosis in CRC. For instance, *IMCA* reduced the viability of CRC cells in vitro and inhibited tumor growth in vivo. *IMCA* also significantly induced ferroptosis in CRC cells, downregulated *SLC7A11* expression, and decreased cysteine and glutathione levels. Activation of the AMPK/mTOR/p70S6k pathway was also found to be related to the activity of *SLC7A11* and ferroptosis [44]. *ACADSB* is weakly expressed in CRC tissues, but it negatively regulates the expression of GPX4 in CRC cells. However, *ACADSB* overexpression inhibits CRC cell migration, invasion, and proliferation. In addition, *ACADSB* overexpression increases the levels of MDA, Fe^+^, superoxide dismutase, and lipid peroxidation in CRC cells but decreases the levels of GSH and GPX4 [45]. *HIF*-2α upregulates lipid and iron regulatory genes in CRC cells and CRC in mice. The activation of *HIF-2α* potentiates ROS production via irreversible cysteine oxidation and enhances cell death. Inhibiting or knocking down *HIF-2α* decreases ROS generation and resistance to oxidative cell death in vitro and in vivo [46].

### 3.3. Non-Coding RNAs Mediate Ferroptosis

In addition to coding genes, studies have reported that non-coding RNAs, including miRNAs, lncRNAs, and circRNAs, also mediate ferroptosis in CRC. Notably, miRNAs generally regulate their target genes in CRC. For example, miR-19a has been identified as an oncogenic miRNA that promotes the proliferation, migration, and invasion of CRC cells. Studies have shown that miR-19a suppresses ferroptosis by inhibiting *IREB2* [47]. In vitro and in vivo experiments indicated that the knockdown of miR-545 enhanced ferroptosis in CRC cells via transferrin regulation. Additionally, miR-539 activates the SAPK/JNK pathway to downregulate the expression of GPX4 and promote ferroptosis by regulating TIPE expression [48]. 

Aside from miRNAs, another study indicated that lncRNAs and circRNAs act as sponges of miRNAs to regulate the expression of downstream genes [49]. LINC01606 has been shown to promote CRC cell growth and invasion both in vitro and in vivo. LINC01606 inhibits ferroptosis by forming a positive feedback regulatory loop with Wnt/β-catenin and decreasing the levels of iron, lipid ROS, and mitochondrial superoxide, and increasing mitochondrial membrane potential [50]. Circ_0007142 was shown to be overexpressed in CRC cells. The inhibition of circ_0007142 facilitated ferroptosis in CRC cells by acting as a sponge for miR-874-3p to upregulate *GDPD5* expression [51]. The knockdown of circABCB10 promoted CRC cell ferroptosis in vitro and inhibited tumor growth in vivo. Mechanistically, circABCB10 sponges miR-326 to regulate *CCL5* expression in CRC cells [52]. Taken together, these studies indicate that both coding and non-coding RNAs mediate ferroptosis in CRC, potentially serving as therapeutic targets in CRC by regulating ferroptosis. 

### 3.4. KRAS Mutation and Ferroptosis in CRC

*RAS* mutations, especially *KRAS* mutation, limit the effectiveness of anti-epidermal growth factor receptor monoclonal antibodies in combination with chemotherapy for metastatic CRC patients. Emerging evidence has shown that ferroptosis is closely related to *KRAS* mutant cells. There were studies showed that mutant *KRAS* dictates a dependency on synthesized fatty acid to escape ferroptosis, establishing a targetable vulnerability in *KRAS*-mutant lung cancer [53]. Pancreatic cancer cells with mutated *RAS* show high levels of glutaminolysis and are thus more susceptible to ferroptosis [54]. However, current knowledge of the correlation between *KRAS* mutation and ferroptosis in CRC is limited. Studies have shown that Cetuximab, β-elemene or Bromelain promoted ferroptosis in *KRAS* mutant CRC cells, which is via inhibition of Nrf2/HO-1 pathway or epithelial-mesenchymal transformation [55,56,57]. These results indicated that induction of ferroptosis could be used to treat *KRAS* mutant CRC cells, thus improving the effectiveness of CRC patients with *KRAS* mutation.

## 4. Agents That Induce or Inhibit Ferroptosis

### 4.1. Plant-Derived Small-Molecule Compounds

Ginsenoside Rh4 (Rh4) is a rare triol ginsenoside that is more soluble in water than other polysaccharide ginsenosides [58]. Rh4 induces proteolysis, apoptosis, and autophagy. In vitro and in vivo experiments have shown that Rh4 triggers ferroptosis by activating autophagy in CRC cells by inducing ROS accumulation [59]. β-elemonic acid (EA) is a triterpene known for its anti-inflammatory and anti-cancer properties. High concentrations of EA (>15 μg/mL) can cause ferroptosis by downregulating the expression of ferritin and upregulating the expression of transferrin, ferroxidase, and ACSL4 [60]. Tetrahydrobiopterin (BH4) is an obligate cofactor of nitric oxide (NO) synthases that plays a redox role in the catalysis of NO formation from L-arginine, O_2_, and NADPH. GTP cyclohydrolase-1 (GCH1)/BH4 prevents lipid peroxidation damage during ferroptosis, which is parallel to the GPX4 redox system. GCH1/BH4 metabolism inhibits ferroptosis by inhibiting NCOA4-mediated ferritinophagy [61]. Auriculasin, isolated from *Flemingia philippinensis,* promoted ROS generation in a concentration-dependent manner. Auriculasin promotes CRC cell apoptosis, ferroptosis, and apoptosis by inducing ROS generation, thereby inhibiting cell viability, invasion, and clone formation [62]. Punicic acid (PunA), a conjugated linolenic acid isomer (C18:3 c9t11c13), has been shown to exert anti-cancer effects. PunA treatment increased intracellular lipid peroxidation. A combination of docosahexaenoic acid synergistically increased the cytotoxicity of PunA in CRC cells [63]. Tagitinin C is a sesquiterpene lactone widely found in plants of the family Asteraceae. Tagitinin C induces ferroptosis through the PERK-Nrf2-HO-1 signaling pathway in CRC cells [64]. Andrographolide is an important diterpenoid lactone isolated from the traditional herb *Andrographis paniculata*. Andrographis acts synergistically with oligomeric proanthocyanidins by activating metabolic and ferroptosis pathways in CRC [65]. Bromelain suppresses Kras-mutant CRC by stimulating ferroptosis [57]. Moreover, Betulaceae extract induces HO-1 expression and ferroptosis in CRC cells [66]. Avicequinone B, a natural naphthoquinone isolated from the mangrove tree, *Avicennia alba*, is a valuable synthetic precursor with an antiproliferative effect. Avicequinone B was shown to reduce the viability of CRC cells and induce G2/M arrest and necrosis-like cell death [67]. (Table 1).

### 4.2. Other Small-Molecule Compounds

Propofol is a commonly used intravenous anesthetic agent that suppresses the proliferation of various human cancers [68]. Propofol induces ferroptosis in CRC cells by downregulating STAT3 and inhibiting GPX4 expression, thereby leading to ferroptosis [69]. Apatinib is a new oral small-molecule agent with anti-angiogenic effects used to treat multiple solid tumors, including CRC. Apatinib promotes ferroptosis in CRC cells by targeting ELOVL6/ACSL4 signaling [70]. Talaroconvolutin A (TalaA) is a natural product isolated from the endophytic fungus *Talaromyces purpureogenus*, which inhabits *Panax notoginseng*. TalaA downregulates the expression of *SLC7A11* and upregulates *ALOXE3* expression to promote ferroptosis in CRC [71]. Dichloroacetate (DCA) is a synthetic, halogenated organic acid. DCA attenuates the stemness of CRC cells by triggering ferroptosis induced by sequestering iron in the lysosomes [72]. BSO is a synthetic amino acid. It irreversibly inhibits gamma-glutamylcysteine synthase, thereby depleting glutathione and resulting in free-radical-induced apoptosis. The cell viability-reducing effects of BSO were attenuated by ferroptosis inhibition and enhanced by iron, indicating that BSO induces ferroptosis in cancer cells [73]. A high-fat diet (HFD) aggravates colitis-associated carcinogenesis by evading ferroptosis through the endoplasmic reticulum (ER) stress-mediated pathway [74]. These lines of evidence demonstrate that many small-molecule compounds can induce or inhibit ferroptosis in CRC, thereby potentially acting as therapeutic agents in CRC (Table 1).

**Table 1 molecules-28-00243-t001:** Agents that induce or inhibit ferroptosis in CRC.

Agents	Target	Reference
**Plant-derived small-molecule compounds**		
Ginsenoside Rh4	ROS generation	[58]
β-Elemonic acid (EA)	transferrin, ferroxidase, *ACSL4*	[60]
Tetrahydrobiopterin (BH4)	*NCOA4*, GPX4	[61]
Auriculasin	ROS generation	[62]
punicic acid (PunA)	MDA, lipid peroxidation	[63]
Tagitinin C	PERK-Nrf2-HO-1 pathway	[64]
Andrographis	*HMOX1*, *GCLC*, *GCLM*, *TCF7L2*	[65]
Bromelain	*ACSL4*	[57]
Betulaceae Extract	*HO*-1	[66]
Avicequinone B	JAK-STAT, MAPK, PI3K-AKT pathway	[67]
**Other small-molecule compounds**		
Propofol	GPX4	[69]
Apatinib	*ELOVL6*/*ACSL4*	[70]
Talaroconvolutin A	*SLC7A11*, *ALOXE3*	[71]
Dichloroacetate	Iron levels	[72]
BSO	GSH	[73]
High-fat diet	*CHAC1*	[74]

### 4.3. Molecules That Mediate Drug Resistance in CRC by Suppressing Ferroptosis

Patients with intermediate- and advanced-stage CRC should be treated with a comprehensive therapeutic regimen, including surgical resection and chemotherapy. Currently, the first-line chemotherapeutic regimens for CRC are FOLFIRI or FOLFOX [75]; however, the major obstacle that affects the prognosis of patients treated with this regimen is chemoresistance. Chemotherapeutic drug resistance often stems from a small pool of mutant cells that acquire selection and growth advantages during treatment [76]. Compared to normal cells, tumor cells are more sensitive to high levels of intracellular ROS; excessive intracellular ROS levels can induce apoptosis in cancer cells. When tumor cells are exposed to chemotherapeutic drugs, they can induce a large amount of ROS; the excessive accumulation of ROS reduces the survival of tumor cells [77]. With the growing knowledge regarding ferroptosis, regulating chemotherapeutic drug resistance by regulating ferroptosis sensitivity has been found to be a promising treatment approach for patients with chemoresistance. Several compounds have been shown to increase ferroptosis sensitivity in drug-resistant CRC cells. 

Andrographolide, a bicyclic diterpenoid lactone, is the principal active component of the medicinal herb *Andrographis paniculata*, a member of the Acanthaceae family. The combination of andrographis with 5-fluorouracil (5-FU) significantly improved its treatment effect, which was orchestrated in part through the dysregulated expression of key genes within the ferroptosis and Wnt signaling pathways [78]. β-elemene, a bioactive compound isolated from the Chinese herb *Curcumae Rhizoma*, exhibits a spectral anti-cancer effect and is used to treat various cancer types. In vivo and in vitro experiments reported that the combination of β-elemene with cetuximab induced ferroptosis and inhibited epithelial-mesenchymal transformation in *KRAS*-mutant CRC cells [56]. Vitamin C (VitC) is an antioxidant that can paradoxically trigger oxidative stress at pharmacological doses. The addition of VitC to cetuximab impaired drug resistance in CRC cells, effectively limiting the onset of acquired resistance to anti-EGFR antibodies [79]. 

In addition to compounds that increase the sensitivity of drug-resistant CRC cells to ferroptosis, some genes were shown to mediate the acquisition of drug resistance in CRC cells and could be potential therapeutic targets. Drug-resistant CRC cells were more sensitive to and underwent ferroptosis induced by GPX4 inhibitors. The strategy of GPX4 inhibition combined with chemotherapy or targeted therapy might be a promising therapy for CRC [80]. A study showed that *FAM98A* inhibits ferroptosis in CRC cells by activating the translation of xCT in stress granules and promoting resistance to 5-FU in CRC by suppressing ferroptosis [81]. Phosphorylated NFS1 weakens oxaliplatin-based chemosensitivity in CRC by preventing apoptosis (apoptosis, necroptosis, pyroptosis, and ferroptosis), increasing intracellular ROS levels [82]. Lipocalin 2 overexpression has been shown to lead to 5-FU resistance in CRC cell lines in vitro and in vivo and inhibit ferroptosis. The underlying mechanism for this was through the decrease in intracellular iron levels and an increase in the expression of GPX4 and a component of the cysteine glutamate antiporter [83]. Moreover, suppressing the KIF20A/NUAK1/Nrf2/GPX4 signaling pathway induces ferroptosis and enhances the sensitivity of CRC cells to oxaliplatin [84].

### 4.4. Nanomaterials That Inhibit CRC by Inducing Ferroptosis 

Since inducing ferroptosis in cancer cells could be used as a novel strategy for the treatment of cancers, various ferroptosis inducers have been identified or developed for this purpose. Unlike traditional therapeutic agents that directly use small molecules, nanotechnology offers new possibilities for triggering ferroptosis for cancer treatment. Treatment effects could be increased and the incidence of side effects could be decreased due to the unique physicochemical properties of nanomaterials or nanoplatforms [85,86]. Li et al. [87] designed a glycyrrhetinic acid-based nanoplatform as a new ICD inducer (GCMNPs), which works in combination with ferumoxytol to promote the Fenton reaction and induce ferroptosis. This nanoplatform was based on boron and nitrogen co-doped graphdiyne, which exhibits efficient GSH/GSH/GPX4 depletion. The combination of GCMNPs and ferumoxytol enhanced the inhibition of PD-1/PD-L1 to activate T cells, subsequently generating a systemic immune response in CRC. DHA also induces ferroptosis in cancer cells. Han et al. [41] developed ZnP@DHA/pyro-Fe core-shell nanoparticles, which stabilized DHA against hydrolysis and prolonged the blood circulation of Chol-DHA and Pyro-Fe for their enhanced uptake in tumors. The co-delivery of an exogenous iron complex and DHA increased ROS production and caused significant tumor inhibition in vivo. These nanoparticles also sensitized non-immunogenic CRC cells to ICD immunotherapy. Pan et al. [88] developed an H2S-responsive nanoplatform based on zinc oxide-coated virus-like silica nanoparticles (VZnO) for the treatment of CRC. VZnO reduces H2S levels in CRC cells and subsequently leads to tumor growth inhibition by activating ferroptosis. More importantly, biosafety-related toxicological and pathological analyses confirmed the low toxicity and great safety of VZnO in CRC treatment. Another newly developed nanoparticle for the deletion of H2S is the FeOOH NSs nanosystem [89]. The FeOOH NSs nanosystem was reported to effectively scavenge endogenous H2S via a reduction reaction to inhibit the growth of CT26 colon cancer cells. The cascade produces FeS driven by H2S overexpression, exhibiting near-infrared-triggered photothermal therapy and Fe^2+^-mediated ferroptosis capabilities. Furthermore, these H2S-responsive nanotheranostics do not cause curative effects on other cancer types. These results highlight the feasibility and characteristics of nanoplatforms and nanomaterials in triggering ferroptosis for CRC therapy. 

Taken together, many genes are related to ferroptosis and several agents could induce ferroptosis in CRC. We believed that inducing ferroptosis by agents or enhancing ferroptosis sensitivity by regulating gene expression or nanomaterials on CRC could be a promising strategy for CRC patients, which serves important clinical application of ferroptosis in CRC. 

### 4.5. Gene Signatures or Clusters That Could Predict the Prognosis of CRC

To date, an increasing number of genes, including coding and non-coding RNA, have been found to be associated with CRC-related ferroptosis, termed ferroptosis-related genes (FRGs). A ferroptosis database was developed to summarize the effects of FRGs [90]. Although some of these genes have been shown to be associated with cancer pathogenesis, development, progression, treatment response, and prognosis, their role in CRC remains to be elucidated. With the rapid development of high-throughput technologies, such as microarrays and RNA sequencing, numerous datasets have been submitted to public databases and can be downloaded freely. Using bioinformatics methods, several researchers have used public datasets to identify gene signatures or clusters to predict CRC prognosis. Table 2 lists the studies reporting the gene signatures or clusters that could predict CRC prognosis. These gene signatures or clusters can be classified into two groups based on their molecular types: mRNAs and lncRNAs. The number of molecules in each signature or cluster varies greatly, typically between 3 and 15 molecules. These signatures or clusters have been reported to show good predictive performance for the prognosis of patients with CRC. CRC is not a single type of tumor; its pathogenesis depends on the anatomical location of the tumor and differs between right-side and left-side of the colon. Left-sided CRCs benefit more from adjuvant chemotherapy regimes and targeted therapies; right-sided CRCs do not respond well to conventional chemotherapies, but they show promising results with immunotherapies [91]. There have also been studies showing gene signatures or clusters with different predictive values for left- and right-sided CRC [92], indicating that molecular differences exist between left- and right-sided CRC, which means that the effect of treatment on CRC patients would greatly improve by targeting specific ferroptosis-related genes based on the location of the tumor. However, most of these studies lack functional validation experiments with individual molecular or clinically independent cohorts to verify their predictive value. Therefore, the results of these studies need to be validated in the future. 

## 5. Conclusions

Reversing treatment resistance in patients with advanced-stage CRC remains a huge challenge for clinicians. A new discovery that regulates cell death could enhance the treatment and prognosis of these patients. The increased sensitivity of CRC cells to ferroptosis via target genes, inducers, or nanomaterials seems to be a feasible strategy to improve the treatment effect of these regimens. However, although substantial progress has been made in understanding the oncogenic states that drive sensitivity to ferroptosis in CRC, considering the complex molecular mechanisms underlying ferroptosis, continuous research is warranted to further explore the interplay of ferroptosis with the genes or inducers in CRC. Moreover, the effects of currently reported inducers need to be further validated in a pre-clinical setting. Collectively, the current progress regarding ferroptosis in CRC revealed an in-depth understanding of the molecular mechanism of this phenomenon and provided candidate treatment targets for CRC. We believe that the combination of ferroptosis inducers or enhancement of ferroptosis sensitivity of CRC cells with chemotherapy, immunotherapy, and radiotherapy will provide an opportunity for new therapeutic approaches in CRC.

## Figures and Tables

**Figure 1 molecules-28-00243-f001:**
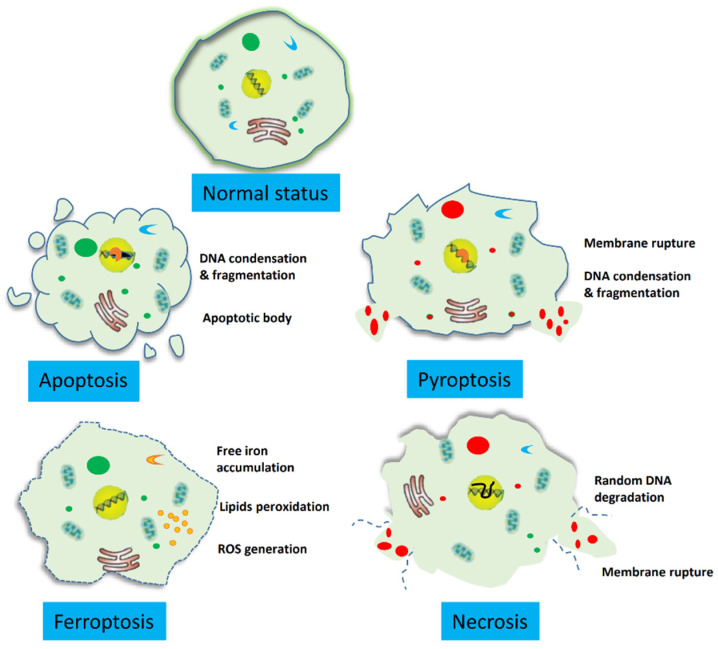
The main morphologic characteristics of apoptosis are pyroptosis, ferroptosis, and Necrosis. Apoptosis is characterized by DNA condensation and fragmentation, and the occurrence of apoptotic body. Cells with pyroptosis present DNA condensation and fragmentation, and the membrane is ruptured. Ferroptosis is defined as free iron accumulation, lipids peroxidation, and ROS generation. Cells undergoing necrosis show DNA degradation and membrane rupture. Pyroptosis and necrosis are accompanied with cell membrane rupture and severe inflammatory reaction, while apoptosis and ferroptosis are devoid of these changes.

**Figure 2 molecules-28-00243-f002:**
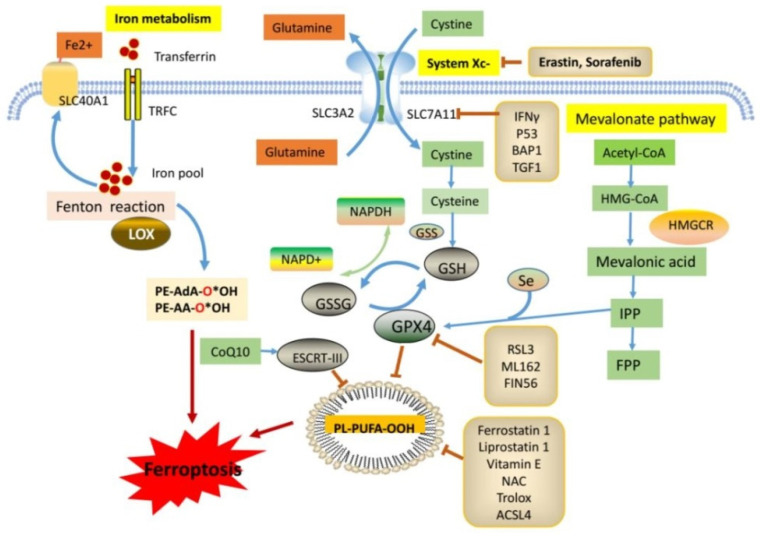
Molecular regulation mechanism of ferroptosis. Three main regulatory pathways govern ferroptosis development: the iron metabolic pathway, system Xc–/GSH/GPX4 pathway, and lipid metabolic pathway. Iron metabolic pathway includes Fe^2+^, transferrin, LOX, and then triggers Fenton reaction to induce ferroptosis. Activation of Xc–/GSH/GPX4 system protecting cells from ferroptosis, which is via reducing the generation of ROS and the following PL-PUFA-OOH. Lipid metabolic pathway involving in Acetyl-CoA-mediated fatty acid synthesis, and induces the accumulation of intracellular free fatty acids to trigger ferroptosis.GPX4: Glutathione peroxidase 4; GSH: Glutathione; LOX: Lipoxygenase; PUFAs: Polyunsaturated fatty acids; ROS: Reactive oxygen species; PL-OOH: phospholipid hydroperoxides. Arrow indicates promoted ferroptosis. Bar represents inhibited ferroptosis.

**Table 2 molecules-28-00243-t002:** Signatures or clusters that could predict the prognosis of CRC.

Type of RNA	Component of Gene Signature or Clusters	Reference
lncRNA	AL161729.4, AC010973.2, CCDC144NL-AS1, AC009549.1, LINC01857, AP003555.1, AC099850.3, AC008494.3	[85]
lncRNA	ZEB1-AS1, LINC01011, AC005261.3, LINC01063, LINC02381, ELFN1-AS1, AC009283.1, LINC02361, AC105219.1, AC002310.1, AL590483.1, MIR4435-2HG, NKILA, AC021054.1, AL450326.1	[93]
LncRNA	AP003555.1, AC099850.3, AL031985.3, LINC01857, STPG3-AS1, AL137782.1, AC124067.4, AC012313.5, AC083900.1, AC010973.2, ALMS1-IT1, AC013652.1, AC133540.1, AP006621.2, AC018653.3	[94]
LncRNA	VCAN-AS1, OVAAL, AC105383.1, AC063952.1, AC129507.1, ITGB1-DT, C15orf54, AC018781.1, NDST1-AS1, AC090204.1, AC011352.1, FAM239A, LINC01210, AC130324.2, LINC01775, AC093458.1, AL022316.1	[95]
LncRNA	AC104819.3, AP003555.1, AC005841.1, LINC02381	[96]
LncRNA	LINC01503, AC004687.1, AC010973.2, AP001189.3, ARRDC1-AS1, OIP5-AS1, NCK1-DT	[97]
Gene	*ACACA*, *GSS*, *NFS1*	[98]
Gene	*AKR1C1*, *ALOX12*, *ATP5MC3*, *CARS1*, *HMGCR*, *CRYAB*, *FDFT1*, and *PHKG2*	[99]
Gene	*NOX4*, *SCP2*, *CARS1*, *ULK1*, *WIPI1*, *CDKN2A*, *BRD4*, *DRD4*, *SLC2A3*, *TFAP2C*	[100]
Gene	*NOS2* and *IFNG* for LCRC; *NOS2* and *ALOXE* for RCRC	[92]
Gene	Two ferroptosis-gene clusters	[101]
Gene	Two ferroptosis-gene clusters	[102]
Gene	Three ferroptosis clusters (FAC1, FAC2 and FAC3)	[103]
Gene	Four subtypes of CRC (C1, C2, C3, C4)	[104]

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
