# Peer review of "Insights on Ferroptosis and Colorectal Cancer: Progress and Updates"

_molecules, 2022, doi:10.3390/molecules28010243_

Round 1

Reviewer 1 Report

Patients with advanced-stage or treatment-resistant CRC benefit less from traditional therapies.  They indicated that the induction of ferroptosis become a new therapeutic approach for CRC.  They reviewed the mechanism of ferroptosis and summarized the value on CRC

The paper has new findings and useful.

Author Response

Patients with advanced-stage or treatment-resistant CRC benefit less from traditional therapies.  They indicated that the induction of ferroptosis become a new therapeutic approach for CRC.  They reviewed the mechanism of ferroptosis and summarized the value on CRC

Point 1: The paper has new findings and useful.

Response: We are highly appreciated your valuable comments.

Reviewer 2 Report

Colorectal cancer (CRC), especially advanced or drug-resistant CRC, is a serious threat to human health, and the induction of ferroptosis may have potential application in the treatment of drug-resistant CRC. Firstly, this paper systematically summarize the generation and regulation of ferroptosis. Secondly, various ferroptosis-related genes that may mediate CRC, as well as the small molecules and nanomaterials as ferroptosis inducers or inhibitors were reviewed. Finally, the authors discussed the clusters of ferroptosis-relevant genes that can be used to predict CRC prognosis. In short, this review summarizes the recent progress on the potential application of ferroptosis in the treatment of CRC. In general, the review focuses on an interesting and clinically important topic and is  overall well-written.

(1) In the Introduction section, the authors should summarize and highlight the relationship between Fe and ROS in ferroptosis, such as the Fenton reaction.

(2) It is better to have an paragraph to compare ferroptosis to other types of cell death, such as apoptosis, pyroptosis, and necrosis. 

(3) Figure 1. More detail description in figure legend should be included. 

(4) Line 98-131: All genes, such as SFRS9, HSPA5, etc., should be italicized.

(5) Line 262: The title should be revised as Nanomaterials that inhibit CRC by inducing ferroptosis. 

(6) Table 1: The title should be revised as Agents that induce or inhibit ferroptosis in CRC. 

(7) Table 2: The title should be revised as Signatures or clusters that could predict the prognosis of CRC. 

Author Response

Colorectal cancer (CRC), especially advanced or drug-resistant CRC, is a serious threat to human health, and the induction of ferroptosis may have potential application in the treatment of drug-resistant CRC. Firstly, this paper systematically summarize the generation and regulation of ferroptosis. Secondly, various ferroptosis-related genes that may mediate CRC, as well as the small molecules and nanomaterials as ferroptosis inducers or inhibitors were reviewed. Finally, the authors discussed the clusters of ferroptosis-relevant genes that can be used to predict CRC prognosis. In short, this review summarizes the recent progress on the potential application of ferroptosis in the treatment of CRC. In general, the review focuses on an interesting and clinically important topic and is overall well-written.

(1) In the Introduction section, the authors should summarize and highlight the relationship between Fe and ROS in ferroptosis, such as the Fenton reaction.
Response: Thank you. We added the content of the relationship between Fe and ROS in ferroptosis in the manuscript. Letters marked in red in the line 67-72, Page 2. 

(2) It is better to have an paragraph to compare ferroptosis to other types of cell death, such as apoptosis, pyroptosis, and necrosis. 
Response: We compared the morphically change of ferroptosis with apoptosis, pyroptosis, and necrosis. Please see the revised Fig. 1 in the Page 2. 

(3) Figure 1. More detail description in figure legend should be included. 
Response: More detail description in figure legend has been added in the revised manuscript. Letters marked in red in the line105-109, Page 3. 

(4) Line 98-131: All genes, such as “SFRS9”, “HSPA5”, etc., should be italicized.
Response: We have modified the font of all genes into italicized through the whole manuscript. 

(5) Line 262: The title should be revised as “Nanomaterials that inhibit CRC by inducing ferroptosis”. 
Response: We modified this title according to your suggestion. Letters marked in red.

(6) Table 1: The title should be revised as “Agents that induce or inhibit ferroptosis in CRC”. 
Response: We modified the title of Table 1 based on your advice. Letters marked in red.

(7) Table 2: The title should be revised as “Signatures or clusters that could predict the prognosis of CRC”.
Response: We modified this title according to your comment. Letters marked in red.